# High prevalence of extrapulmonary tuberculosis in dairy farms: Evidence for possible gastrointestinal transmission

**Fang Xu[1,2], Lili Tian[2], Yan Li[3], Xuelian Zhang[4]\*, Yayin Qi[5], Zhigang Jing[2], Yangyang Pan[1], Li Zhang[3], Xiaoxu Fan[2], Meng Wang[1], Qiaoying Zeng[1]\*, Weixing Fan[2]\***

**1** College of Veterinary Medicine, Gansu Agricultural University, Lanzhou, Gansu, China, **2** National Animal Tuberculosis Reference Laboratory, Division of Zoonoses Surveillance, China Animal Health and Epidemiology Center, Qingdao, Shandong, China, **3** Animal Husbandry and Veterinary Station, Xinjiang Production and Construction Corps, Urumqi, Xinjiang, China, **4** State Key Laboratory of Genetic Engineering, School of Life Science, Fudan University, Shanghai, China, **5** College of Animal Science and Technology, Shihezi University, Shihezi, Xinjiang, China

\* zengqy@gsau.edu.cn (QZ); fanweixing@cahec.cn (WF); xuelianzhang@fudan.edu.cn (XZ)

**Data Availability Statement:** All relevant data are within the paper and its Supporting Information files.

## Abstract

Bovine tuberculosis (bTB) caused by *Mycobacterium bovis* (*M. bovis*) represents one of major zoonotic diseases among cattle, it also affects the health of human, other domestic animals and wild life populations. Inhalation of infected aerosol droplets is considered as the most frequent route of the infection. This study aims to investigate the current forms of tuberculosis in cattle and identify the possible transmission modes in dairy farms of China. 13,345 cows from eight dairy farms in three provinces were comprehensively diagnosed by a multitude of assays, including SIT, CIT, IFN-γ assay and ELISA. It has been indicated that advanced infection of bTB was found in 752 (5.64%) cattle, suggesting a high prevalence of tuberculosis in these dairy farms. In the necropsy examination of 151 positive cattle, typical bTB lesions were observed in 131 cattle (86.75%), of which, notably, 90.84% lesions appeared in liver, spleen, mesenteric lymph nodes, mammary lymph nodes and other organs, taking up a large proportion among cattle with advanced bTB infection. 71.26% extrapulmonary tuberculosis (EPTB) was related to gastrointestinal system. *M. bovis* nucleic acid was further found in milk and feces samples and *M. bovis* was even isolated from milk samples. Phylogenetic analysis based on whole genome sequencing unraveled that six isolates were closely related to *M. bovis* AF2122/97 originated from UK, whereas four isolates shared close relation to *M. bovis* 30 from China, respectively. Our data demonstrate that the increase of EPTB transmitted by digestive tract is implicated in the current high prevalence rate of bTB in China, which also provides leads for bTB control in other countries with high prevalence of bTB in the future.

## Introduction

Bovine tuberculosis (bTB) is a chronic bacterial disease caused by *Mycobacterium bovis*, which is harmful to humans, livestock and wildlife populations [1, 2]. Presently, bTB remains a major

**Funding:** This work was supported by Gansu Agriculture Research System (GARS-CS-5); China Agriculture Research System (CARS-36); and the National Natural Science Foundation of China (81673482, 81971898).

**Competing interests:** The authors have declared that no competing interests exist.

infectious disease in developing countries, and causes heavy burden in both public health and animal husbandry [2, 3]. bTB is mainly transmitted through the respiratory tract, which can result in typical symptoms, such as cough and dyspnea. Of note, the infection generally leads to the development of characteristic lesions, tuberculous nodular granuloma, and caseous necrosis or calcification in the lungs, pleura and pleural lymph [4]. In fact, digestive tract exposure contributes to another critical mode for tuberculosis transmission, as humans can become infected most commonly through consumption of unpasteurized milk products from infected cows [5]. Importantly, in the dairy farm, cattle can also become infected following ingestion of feed or water contaminated with nasal secretions, feces, urine, or unpasteurized milk from infected animals [4].

Current ante mortem diagnosis of bTB mainly relies on the single intradermal test (SIT) and IFN-γ assay, the comparative intradermal test (CIT) is also adopted alternatively to SIT. However, recent studies revealed that, in positive cattle by skin test, neither typical bTB-associated symptoms, such as cough and dyspnea, nor lung lesions upon necropsy were frequently found [6, 7]. This confusion has somehow even given rise to a loss of confidence among farmers and field veterinarians regarding the accuracy of delayed hypersensitivity test, and it also interfered the implementation of culling policy on the cattle suspected to bTB [6, 7]. Currently, in China, the postmortem examination of PPD positive cattle is often limited to lung and thorax in terms of bio-safety consideration. Additionally, in slaughterhouses, tissues and organs such as liver, spleen, intestine and stomach, are usually removed, the lesions in which are therefore frequently overlooked [8, 9]. It is believed that the route of infection is implicated to the nature, extent and distribution of tuberculous lesions [10]. For instance, oral transmission often leads to the development of primary foci in lymph tissues of the intestinal tract [11], during which, mesenteric lymph nodes become the most susceptible target organs [12]. However, there were still few details on the distribution of extrapulmonary tissues and organs of bTB. In this study, we aim to investigate the EPTB in cattle and seek to find clues for the possible transmission route in dairy farms.

## Materials and methods

### General information of dairy farms

Eight dairy farms from Xinjiang, Shandong and Guangxi provinces were chosen based on the epidemic state of bTB. These farms that raised Holstein cows adopted the intensive livestock production systems. The feeding of cows was based on total mixed ration (TMR) ad libitum. Previous bTB positive rates of these dairy farms were between 5%~15% as determined by twice-yearly SIT testing.

### Skin test

The skin test was carried out in all animals after challenge. SIT was performed by inoculating 0.1 mL (3000 IU) of bovine-PPD (Prionics AG, Thermo Fisher Scientific, USA) in the midneck. CIT was conducted by inoculating 0.1 mL (3000 IU) of bovine-PPD and 0.1 mL (2500 IU) avian-PPD (Prionics AG, Thermo Fisher Scientific, USA) in the neck, and the distance between the two injection points was approximately 15 cm. The skin-fold thickness was measured before and at 72 h after injection. The results of skin thickness increase were interpreted according to the standards of official criteria (GB/T 18645–2002 and OIE Terrestrial Manual 2018) for both SIT and CIT. A cow was considered SIT positive, inconclusive or negative when the increase was 4 mm or greater, between 2 and 4 mm or less than 2 mm, respectively. For CIT, cattle were deemed positive, inconclusive or negative when the bovine injection site

exceeded the avian site by greater than 4, 1~4 and 1 mm or less, respectively. CIT was conducted two months after SIT.

### IFN-γ assay

Whole blood was collected from the jugular or caudal vein in tubes with lithium heparin. Stimulation of whole blood with bovine-PPD, avian-PPD or PBS (nil control) was carried out within 8 h after collection. The BOVIGAM® *Mycobacterium bovis* Gamma Interferon Test Kit for Cattle (Prionics AG, Thermo Fisher Scientific, USA) was used for the detection of the release of IFN-γ from sensitised lymphocytes according to the manufacturer's protocol. The result interpretation was obtained based on the standard cut-off value of the kit.

### ELISA

A commercial ELISA, the *Mycobacterium bovis* Antibody Test Kit (IDEXX, USAhttps://www.idexx.com/), was used to test all serum samples examined in this study. The assay and results interpretation were performed according to the manufacturer's protocol. The results are expressed as a Sample to Positive (S/P) ratio, which is calculated for each sample according to the following equation: S/P = (Mean Sample OD450 − Mean Negative Control OD450) / (Mean Positive Control OD450 − Mean Negative Control OD450). The sample was positive with S/P ≥ 0.30.

### Comprehensive diagnostic algorithm

Four immunological assays were employed to detect bTB in cattle from the eight dairy farms investigated, so as to reduce the false positive and false negative rates. Serological test (ELISA) detecting humoral immune responses can facilitate the detection if late stage diseased animals. In serial testing, positive results from two or more assays (including ELISA) were considered as advanced infection, and anatomical examination was performed in the local slaughterhouses named Kaerwan (Xinjiang), Musulin (Shandong), and Chengcheng (Guangxi). Different tissues and organs were collected for subsequent pathological and pathogenic examination. Milk and feces samples from cattle with advanced infection were detected by bacteria isolation and PCR assay (S1 Fig).

### Anatomical and pathological examination

Anatomical examination was systematically conducted to 151 cattle. Numerous organs and tissues were thoroughly inspected and collected for examination of bTB-compatible lesions, including lung, liver, spleen, kidney, intestine, thoracic cavity and pleura, abdominal cavity and peritoneum, hilar lymph nodes, mammary lymph nodes, mesenteric lymph nodes, inguinal lymph nodes. Tissues were fixed in 10% formalin neutral solution. Conventional paraffin sectioning and H&E staining were conducted for histopathological examination. Furthermore, paraffin sections were stained with Ziehl-Neelsen acid-fast staining for detection of tubercle bacillus.

### Bacteria isolation

*M. bovis* was isolated and cultured at the Biosafety Level-3 Laboratory for Zoonoses, China Animal Health and Epidemiology Center (CAHEC) (Qingdao, Shandong, China). Briefly, approximately 2 g of tissue were collected, homogenized in 15 mL of 0.85% physiological saline and vortexed at high speed for 1 minute. 15 mL of homogenate was transferred to a centrifuge tube, after which 15 mL of hexadecylpyridinium chloride monohydrate (HPC) 1.5% (w/v) was

added and mixed at room temperature for 30 minutes. The mixture was centrifuged at 3,000 × g for 20 minutes at 22˚C, and the pellets were resuspended in 10 mL of 0.85% physiological saline as inoculum. Milk samples were similarly treated. 20 mL of milk were centrifuged at 3,000 × g for 10 minutes, and pellets were decontaminated in 20 mL of HPC 0.75% (w/v) for 20 minutes. After centrifugation and suspension, 100 μL of tissue or milk samples were inoculated on to Lowenstein-Jensen medium supplemented with pyruvate (BD, USA) or Lowenstein-Jensen medium (BD, USA), at 37˚C for 4~10 weeks. The growth of bacteria was monitored weekly.

## DNA extraction and purification

DNAs from all positive cultures, milk, and feces were extracted by QIAamp® DNA Mini Kit (QIAGEN, Germany) and QIAamp® DNA Stool Kit (QIAGEN, Germany). Purified DNA concentrations were determined using a NanoPhotometer® NP80 (Implen, Germany).

## Detection of *M. bovis* by quantitative PCR (qPCR)

The qPCR assay targeting IS*1561* of Mycobacterium tuberculosis complex (MTBC) species was performed. The amplicon was then further analyzed using the RD4 based qPCR assay to confirm the presence of *M. bovis* [13]. Taqman Universal PCR Master Mix was used according to the manufacturer's directions with a final primer concentration of 0.4 μM each and 0.2 μM probe. The qPCR was conducted with Applied Biosystems™ QuantStudio 3 Real-Time PCR Systems (Thermo Fisher Scientific, USA). Initial denaturation was made at 95˚C for 15 min, followed by 45 cycles with denaturation at 95˚C for 15 s, annealing and elongation at 60˚C for 1 min.

## Whole genome sequencing (WGS)

Ten bacterium strains were isolated from eight dairy farms, while six strains from lungs (2016–4, 2017–12, 2017–14, 2017–15, 2017–18) and four strains from extrapulmonary tissues (2016–1, 2016–2, 2016–6, 2017–13, 2017–17). WGS was performed on the BGISEQ-500 platform (BGI, Shenzhen, China). Reads were generated and assembled into contigs, while the sequence of *M. bovis* AF2122/97 (GenBank: NC_002945.4) was set as reference.

We introduced Prokka to fully annotate draft bacteria genomes. All annotated assemblies in GFF3 format were used as input files to conduct a core-pan analysis. SNP sites were extracted from a core gene alignment file generated by Roary. Missing and incomplete data were excluded from analysis. A matrix dataset containing the orthologous SNPs was generated, and the filtered dataset was applied to conduct evolutionary analyses using MEGA version 5. A neighbor-joining tree was constructed using the Jukes–Cantor model and the percentage bootstrap confidence levels of internal branches were calculated from 1,000 resamplings of the original data. The *M. tuberculosis* H37Rv strain (GenBank: NC_000962.3) was selected as the outgroup. Six *M. bovis* strains from NCBI were selected as reference. Specifically, *M. bovis* AF2122/97 (GenBank: NC_002945.4) was isolated from the lung and bronchomediastinal lymph nodes of a cow in the UK, *M. bovis* 1595 (GenBank: NZ_CP012095.1) was isolated from the larynopharyngeal lymph node of a cow in South Korea, *M. bovis* 30 (GenBank: CP010332.1) was isolated from the lymph node of a cow in China, *M. bovis* AN5 (GenBank: NZ_AWPL00000000.1) was obtained from Brazil and used for PPD production, *M. bovis* SP38 (GenBank: NZ_CP015773.2) was isolated from the lymph node of a cow in Brazil, and *M. bovis* BCG str. Tokyo 172 (GenBank: NC_012207.1) was vaccine strain.

### Ethical statement

bTB is a notifiable disease and there are control and surveillance campaigns in China. Official diagnostic methods for bTB are immunological tests, culture, PCR and histopathology. Skin tests and tissue collection were included as part of routine surveillance strategy of bTB. Sample collection and subsequent detection were conducted under the condition of permission of farmer owners. In this study, no animal experiment was involved. All datasets were in complete agreement with national and OIE regulations.

## Results

### High prevalence of tuberculosis in dairy farms

The bTB infection in dairy farms from multiple provinces of China was investigated. In this study, the positive rates from a total of 13,345 cattle from eight dairy farms were determined by using SIT, CIT, IFN-γ assay and ELISA. Our result showed that the positive rates from different dairy farms varied, ranging from 6.15% to 31.38% (SIT), 2.75% to 24.13% (CIT), 3.00% to 24.44% (IFN-γ assay), and 1.60% to 13.94% (ELISA), respectively (Table 1). The average positive rates among 13,345 cattle were 14.79%, 12.25%, 12.28%, 5.55%, by using immunological detection methods. Collectively, 752 (5.64%) cattle were further diagnosed with advanced infection, confirmed by both IFN-γ assay and ELISA, suggesting a high prevalence of tuberculosis in these dairy farms.

### A large proportion of EPTB among cattle with advanced bTB infection

One hundred and fifty one animals from all eight dairy farms were selected and received further anatomical examination. In fact, typical bTB lesions were observed in 131 (86.75%) cattle, among which, 119 cattle (90.84%) were found with lesions regarding EPTB (Fig 1A), including extensive cavitary lesions in liver, tuberculous granuloma in spleen, granulomas with caseous necrosis and mineralization in mammary lymph node, gastric lymph node, mesenteric lymph node, intestinal lymph node, hilar lymph node and lung, miliary generalised tuberculosis on the thoracic cavity pleura (tuberculous "pearls") (Fig 2). H&E staining result further validated typical tuberculous granulomas with evident necrosis and mineralization, numerous epithelioid cells and Langhans' giant cells in the intermediate layer of the granuloma, a large number of lymphocytes in the external layer of the granuloma, as well as caseous necrosis of granulomata (Fig 3). There were bTB lesions associated with gastrointestinal system, which accounted for 71.26% of EPTB (Fig 1B and 1C).

**Table 1. Results of immunological detection.**

| Farm | No. cattle | No. bTB positive cattle (positive rate/%) | | | | No. cattle with advanced infection (positive rate/%) | No. necropsy |
|------|-----------|-------------|-------------|-------------|-------------|------------------------------|---------|
| | | SIT | CIT | IFN-γ assay | ELISA | | |
| A | 2,000 | 123(6.15) | 55(2.75) | 60(3.00) | 32(1.60) | 56(2.80) | 23 |
| B | 1,690 | 280(16.57) | 245(14.50) | 226(13.37) | 65(3.85) | 58(3.43) | 38 |
| C | 1,600 | 502(31.38) | 386(24.13) | 391(24.44) | 223(13.94) | 241(15.06) | 10 |
| D | 1,300 | 95(7.31) | 85(6.54) | 83(6.38) | 71(5.46) | 68(5.23) | 25 |
| E | 2,300 | 359(15.61) | 322(14.00) | 396(17.22) | 111(4.83) | 123(5.35) | 17 |
| F | 1,755 | 412(23.48) | 378(21.54) | 352(20.06) | 89(5.07) | 76(4.33) | 8 |
| G | 1,200 | 79(6.58) | 69(5.75) | 45(3.75) | 44(3.67) | 38(3.17) | 25 |
| H | 1,500 | 124(8.27) | 95(6.33) | 86(5.73) | 106(7.07) | 92(6.13) | 5 |
| Total | 13,345 | 1,974(14.79) | 1,635(12.25) | 1,639(12.28) | 741(5.55) | 752(5.64) | 151 |

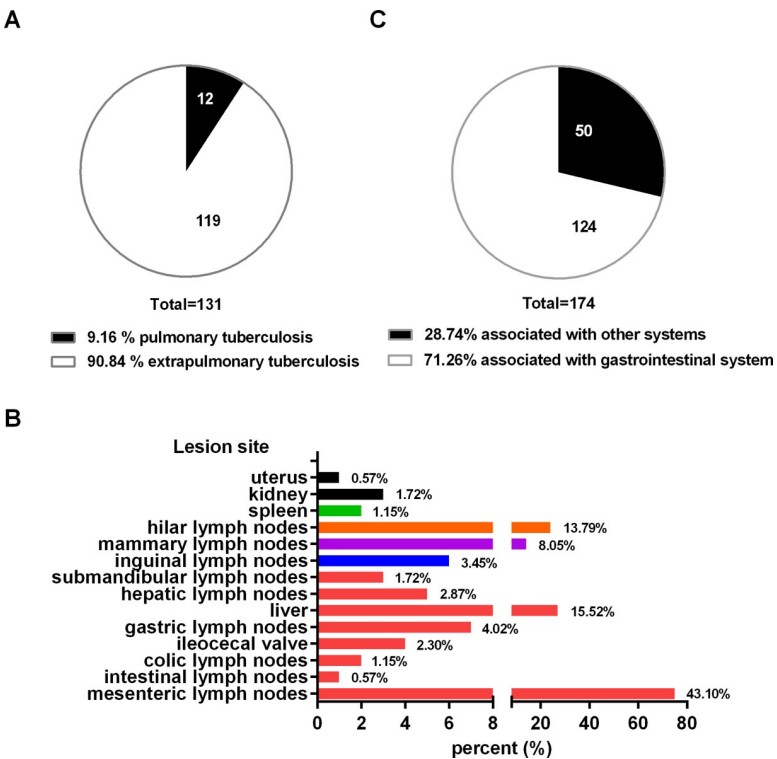

**Fig 1. Lesion distribution.** (A) Proportion of extrapulmonary tuberculosis and pulmonary tuberculosis in slaughtered cattle. (B) Proportion of lesion sites in extrapulmonary tuberculosis. (C) Proportion of lesions associated with gastrointestinal system and other system.

## *M. bovis* isolation and identification

Sections of all paraffin-embedded tissues isolated from 151 cattle were treated with Ziehl-Neelsen acid-fast staining, and red-stained rod-shaped bacteria were observed from samples of 26 cattle (S2 Fig). The isolation of MTBC was then performed from 289 diverse kinds of tissues of 151 cattle. We successfully isolated the bacteria from 113 tissue samples of 91 cattle, accounting for 60.26% (91/151) of the total cattle examined. Bacterial colonies exhibited pellet, tubercle, or cauliflower shapes with a beige and creamy-white color on solid medium (S3 Fig). These bacteria were then identified as *M. bovis* by qPCR (S4 Fig).

As shown in the phylogenetic tree, 17 MTBC strains from different species clustered into two major clades, all 10 isolates in this study belonged to the *M. bovis* clade (Fig 4). Notably, 6 isolates were closely related to *M. bovis* AF2122/97 originated from UK, whereas 4 isolates shared close relation to *M. bovis* 30 from China, indicating the complicated epidemic status of *M. bovis* in dairy farms of China.

## Evidence for possible gastrointestinal transmission

Field investigations at these dairy farms identified that the colostrum and regular milk for calves feed were not fully sterilized through pasteurization. In addition to the large proportion of bTB lesions sites associated with gastrointestinal system among EPTB, we further investigated the possible routes of tuberculosis transmission in these dairy farms. 54 milk samples, including 46 from positive cattle and 8 from calving room, were collected. qPCR result revealed *M. bovis* nucleic acid positive in all 8 samples from calving room and 39 samples from

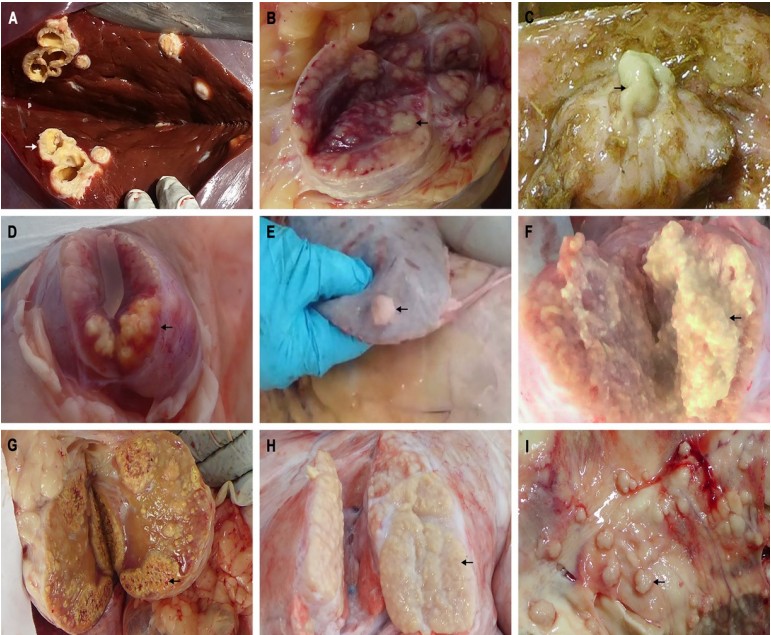

**Fig 2. Lesions observed by necropsy.** (A) Extensive cavitary lesions in liver. (B) Granulomas with caseous necrosis in mammary lymph node. (C) Granulomas with caseous necrosis in gastric lymph node. (D) Granulomas with caseous necrosis and mineralization in mesenteric lymph node. (E) Tuberculous granuloma in spleen. (F) Granulomas with caseous necrosis and mineralization in intestinal lymph node. (G) Granulomas with caseous necrosis and mineralization in hilar lymph node. (H) Big granulomas with caseous necrosis and mineralization in lung. (I) Miliary generalised tuberculosis on the thoracic cavity pleura (tuberculous "pearls").

positive cattle. 12 *M. bovis* isolates were further cultured from 54 milk samples. We also tested the 54 feces samples. Similar to milk samples, *M. bovis* nucleic acid was detected in all 8 samples from fecal pool and 34 feces samples from bTB-positive cattle. However, no *M. bovis* was successfully isolated from these 54 feces samples (Table 2). These results unraveled the possibility of gastrointestinal transmission mode of EPTB in dairy farms.

## Discussion

Bacteriological and histopathological examination of tissues are recognized as the most accurate and reliable methods for tuberculosis detection in cattle [14]. Given that isolation of *M. bovis* preferably requires 10~12 weeks, macroscopic lesions in carcasses at slaughterhouse are of great necessity for rapid detection and prompt actions for disease intervention [15]. So far, tuberculin test has been the standard method for detection of bTB worldwide [16]. For instance, the SIT has been extensively employed in the Irish bTB eradication program and has proven to be a very safe means to test and screen the Irish cattle population [17]. In several dairy farms of China, a high prevalence of bTB is reported by using the same method [18]. The complexity of bTB pathogenesis influences the conclusive presence of infection based on a single detection test, while the specificity and sensitivity may vary among the different immunological methods depending on the stages of infection. The combination of pathological and etiological methods is typically used to confirm *M. bovis* infection. In this scenario, we performed a comprehensive diagnosis by employing four immunological assays, SIT, CIT, IFN-γ assay and ELISA to reduce the false positive and false negative rates, in addition to pathological changes. By screening 13,345 cattle from 8 dairy farms, we found high prevalence of tuberculosis (2.8~15.06% of advanced infection) in these dairy farms. We further isolated and identified

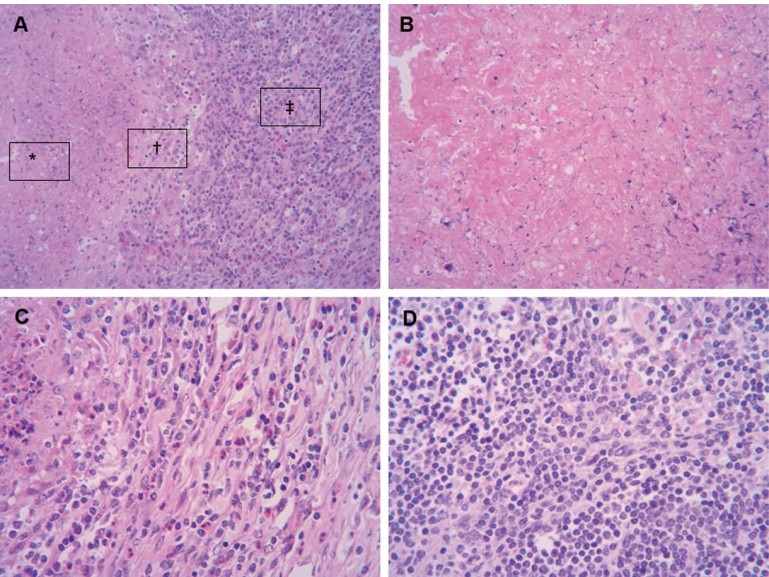

**Fig 3. H&E staining of sections of hilar lymph nodes.** (A) Typical tuberculous granulomas consisted of three parts, as shown at the box: the center of the granuloma displays evident necrosis and mineralization (*); numerous epithelioid cells and Langhans' giant cells are present in the intermediate layer of the granuloma (†); A large number of lymphocytes are present in the external layer of the granuloma (‡). Magnification ×200. Box (*), Box (†), and Box (‡) indicate areas enlarged in panel B, C, and D, respectively. (B) Amorphous pink caseous material composed of the necrotic elements of the granuloma as well as the infectious organisms. Magnification ×400. (C) Epithelioid cells aggregation and Langhans' giant cells could be seen subsequently. Magnification ×400. (D) Redundant lymphocytes infiltrated. Magnification ×400.

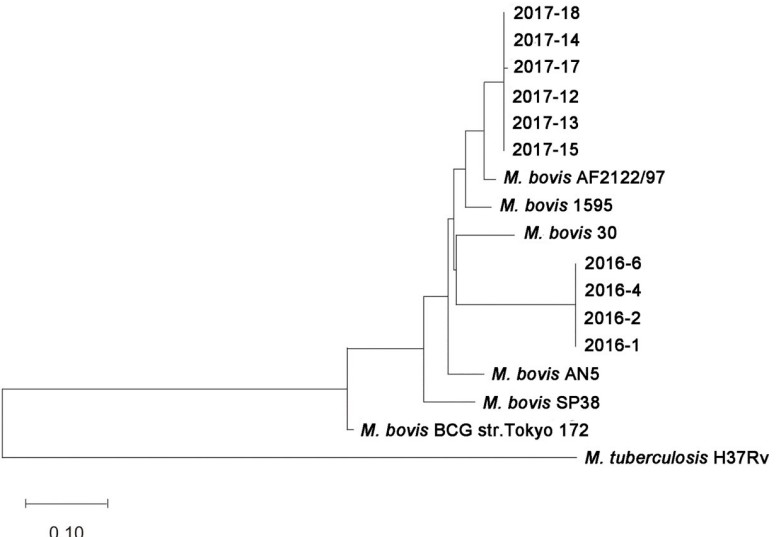

**Fig 4. Phylogenetic trees of ten isolates.** The optimal tree with the sum of branch length = 1.71277593 was shown in Fig 4. The evolutionary distances were calculated using the Maximum Composite Likelihood method and were in the units of the number of base substitutions per site. This analysis involved 17 nucleotide sequences. All ambiguous positions from each sequence pair were removed (pairwise deletion option). There were a total of 3,914 positions in the final dataset. Evolutionary analyses were conducted in MEGA 5.

**Table 2. The results of tissue, milk and feces samples by culture and qPCR.**

| Sample type | Sample source (eight farms) | Number | Culture (+) | qPCR (+) | Ct value |
|---|---|---|---|---|---|
| **Tissue** | Positive cattle | 151 | 91 | 91 | 17.56–30.11 |
| **Milk** | Positive cattle | 46 | 10 | 39 | 30.04–35.59 |
| | Calving room | 8 | 2 | 8 | 31.79–35.13 |
| **Total** | | 54 | 12 | 47 | |
| **Feces** | Positive cattle | 46 | 0 | 34 | 34.85–36.80 |
| | Fecal pool | 8 | 0 | 8 | 34.60–36.07 |
| **Total** | | 54 | 0 | 42 | |

the pathogen and the result of WGS-SNP analysis of ten isolates confirmed they belonged to different origins of *M. bovis* clade.

The previous study on emergence of virulent isolates of *M. bovis* in the Nile Delta present that positive CIT results were identified in 3% of the animals spread among 40% of the examined herds. Post-mortem examination of slaughtered cattle then revealed the presence of both pulmonary and/or digestive forms of tuberculosis in > 50% of the examined animals [19]. Another finding regarding badgers naturally infected with *M. bovis* exhibited that most transmission occurred by the respiratory route, due to predominance of lesions in the respiratory tract [20]. The evidence on naturally infected dromedary reported that a total of 18 (19.56%) camels out of 92 examined revealed two different TB-lesion patterns, pulmonary (n = 15) and disseminated (n = 3) forms, suggesting that the pulmonary form of the TB was more common in camels [21]. Intriguingly, based on experimental infection studies with *M. bovis* that the transmission route affects the distribution of tuberculose focus and tissue tropism [10], our postmortem finding revealed that the majority of bTB occurred was EPTB (90.84%), as lesions in the lung were only detected among 9.16% of infected cattle in this study, suggesting EPTB as the major manifestation of infection in these dairy farms. This result was in accordance with previous finding in Ethiopia that the majority of bTB lesions located in the mesenteric lymph nodes, and the frequency and severity of the lesions were higher in the mesenteric lymph nodes than the thoracic lymph nodes [22]. They suggested that shedding of *M. bovis* in the feces and ingestion of the bacilli from contaminated pasturage and/or water may be the main route of transmission in pasture cattle, as lesions were primarily observed in mesenteric lymph nodes. Similarly, our data suggests that the gastrointestinal tract lesions in EPTB were initial foci, for *M. bovis* was found in colostrum and regular milk and feces.

In developed countries, effective control and eradication strategies include sterilization of milk by strict pasteurization procedures. In this scenario, lesions are mainly found in the lungs and lymph nodes in most of sporadic cases, suggesting the primary transmission route via respiratory tract [23]. It is therefore believed that milk, urine and feces play a minor role in transmission of bTB [24]. However, there are few cases reported regarding gastrointestinal tract infection with mesenteric lymph node lesions and intestinal wall tuberculosis nodules, the cause of which is related to contaminated pasture and drinking water by wildlife [25]. In these reports gastrointestinal tract lesions were far less common than respiratory tract lesions [26]. Nevertheless, our studies indicate that EPTB was mainly found among PPD-positive cattle in dairy farms, especially tuberculosis in digestive tract. We speculate that calves were mainly infected by drinking colostrum or regular milk that were not completely sterilized through pasteurization, or by drinking bacteria-contaminated forage or water in the dairy farms. It is proposed that the gastrointestinal transmission mode may lead to high prevalence of bTB in herds, on account of inadequate pasteurization and wildlife reservoirs in many developing countries [27]. Our preliminary data indicate the presence of EPTB in cattle farms

throughout multiple provinces of China, owing to transmission via digestive tract. However, the limitation in this study still exists that the high prevalence rate of bTB in cattle farms caused by gastrointestinal transmission requires to be further investigated throughout China.

## Conclusions

In conclusion, our data demonstrate that the bovine EPTB is the major manifestation of bTB infection in dairy farms, linking the evidence of oral transmission route. It attaches the attention on pasteurization programs in *M. bovis*–prevalent areas to restrict possible transmission of bTB through the consumption of dairy products.

## Supporting information

**S1 Fig. Comprehensive diagnosis of bovine tuberculosis in dairy farms.** SIT, single intradermal test; CIT, comparative intradermal test; IFN-γ, gamma-interferon; ELISA, enzyme-linked immunosorbent assay; PCR, polymerase chain reaction.
(TIF)

**S2 Fig. Ziehl-Neelsen acid-fast staining of mesenteric lymph node.** Sections were stained by Ziehl-Neelsen acid-fast staining and images were captured and shown at ×400. Acid fast bacilli were stained with red and were observed under microscope.
(TIF)

**S3 Fig. Isolation and culture of *M. bovis*.** (A) Creamy-white pellet colony. (B) Beige granular colony. (C) Nodular colony. (D) Cauliflower like colony.
(TIF)

**S4 Fig. The results of qPCR.**
(TIF)

## Acknowledgments

We thank Yanlin Zhao, Xichao Ou, Bing Zhao, Guangxian Xu and Zhanbing Ma for their technical assistance in conducting this research.

## Author Contributions

**Conceptualization:** Fang Xu, Weixing Fan.

**Data curation:** Fang Xu.

**Formal analysis:** Fang Xu, Lili Tian, Zhigang Jing.

**Funding acquisition:** Xuelian Zhang, Qiaoying Zeng, Weixing Fan.

**Investigation:** Fang Xu, Lili Tian, Yan Li.

**Methodology:** Fang Xu, Weixing Fan.

**Project administration:** Qiaoying Zeng, Weixing Fan.

**Resources:** Fang Xu, Lili Tian, Yan Li, Yayin Qi, Zhigang Jing, Li Zhang, Meng Wang.

**Software:** Fang Xu, Zhigang Jing.

**Supervision:** Qiaoying Zeng.

**Validation:** Fang Xu.

**Visualization:** Fang Xu, Xuelian Zhang.

**Writing – original draft:** Fang Xu.

**Writing – review & editing:** Xuelian Zhang, Yangyang Pan, Xiaoxu Fan, Qiaoying Zeng, Weixing Fan.

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
