## [Decision Letter · Decision Letter 0]

18 Aug 2020

PONE-D-20-23226

High prevalence of extrapulmonary tuberculosis in dairy farms: evidence for possible gastrointestinal transmission

PLOS ONE

Dear Dr. Zeng,

Thank you for submitting your manuscript to PLOS ONE. After careful consideration, we feel that it has merit but does not fully meet PLOS ONE’s publication criteria as it currently stands. Therefore, we invite you to submit a revised version of the manuscript that addresses the points raised during the review process.

Please submit your revised manuscript. If you will need more time than this to complete your revisions, please reply to this message or contact the journal office at plosone@plos.org. Please include the following items when submitting your revised manuscript:

We look forward to receiving your revised manuscript.

Kind regards,

Frederick Quinn

Academic Editor

PLOS ONE

Journal Requirements:

2. In your Methods section, please provide the name of the slaughterhouse where the animals were sacrificed.

3. Thank you for including your ethics statement:  "All ante-mortem testing and post-mortem samples were obtained in compliance with national standards of the People's Republic of China (PRC): Diagnostic techniques for tuberculosis of animal (GB/T 18645-2002), and Bovine tuberculosis diagnosis-Assay on IFN-γ in vitro (GB/T 32945-2016); in accordance with national guidelines for the prevention and treatment of bovine tuberculosis (2017-2020) (Ministry of Agriculture and Rural Affairs of PRC); in accordance with legislation: Animal Epidemic Prevention Law of PRC (1997, order No.87 of president of PRC).".   

Please amend your current ethics statement to confirm that your named ethics committee specifically approved this study.

For additional information about PLOS ONE submissions requirements for ethics oversight of animal work, please refer to http://journals.plos.org/plosone/s/submission-guidelines#loc-animal-research  

Reviewers' comments:

Reviewer's Responses to Questions

**Comments to the Author**

1. Is the manuscript technically sound, and do the data support the conclusions?

Reviewer #1: Partly

Reviewer #2: Yes

Reviewer #3: No

2. Has the statistical analysis been performed appropriately and rigorously? 

Reviewer #1: No

Reviewer #2: Yes

Reviewer #3: Yes

3. Have the authors made all data underlying the findings in their manuscript fully available?

Reviewer #1: Yes

Reviewer #2: No

Reviewer #3: Yes

4. Is the manuscript presented in an intelligible fashion and written in standard English?

Reviewer #1: No

Reviewer #2: No

Reviewer #3: No

5. Review Comments to the Author

Reviewer #1: The current paper investigates extra-pulmonary tuberculosis in dairy farms. 8 farms have been included in the study with a total of 13345 cattle in 3 different provinces in China. bTB diagnostic was performed using SIT,  CIT, IFN-γ assay, PPD eye drop reaction and ELISA. In addition 151 positive cattle were examined by necropsy and revealed 71.26% of the lesions localized in the liver, spleen, mesenteric lymph nodes, mammary lymph nodes and other organs and were classified as extra-pulmonary TB, while only 9.16% of the animals had lesions in the lungs.

The study discusses the possibility of transmission of bTB through consumption of infected colostrum, regular milk, forage and water rather than aerosol transmission, since a small proportion of the animals had lesions in their lungs.

Overall, the manuscript present interesting data but needs to be improved by a in-depth data analysis and more exhaustive discussion, suggestions are included below, in addition to some questions:

Please explain from an immunological point of view how did you chose the different tests to design the early infection versus the advanced infection.

I suggest you add the methods of each test briefly and then reference the instructions given by the vendor when appropriate. The same for the qPCR, it will be good to add a brief description of the experiment

In the methods section, I suggest you add a description of the livestock production systems (intensive or extensive) of the farms included in the study, in addition to a brief description of the animal housing conditions and the breeds of the cattle screened. Those are all elements that can influence the transmission and persistence of bTB in livestock.

Table 1 gives a great summary of the results of bTB diagnosis, however, it will be interesting to perform a multivariate analysis in order to detect any significant statistical difference between the different methods used.

In addition, it will be useful to calculate the apparent and true prevalence considering the sensitivity and specificity of each test used (or the tests for which sensitivity and specificity have been previously determined)

I suggest to add some information to the introduction part, some questions to answer in the introduction:

- is the comparative tuberculin skin test using avian PPD in addition to bovine PPD used in China

- Are there any available molecular data of bTB in cattle in China (pulmonary and extra pulmonary)

- In Slaughterhouses, usually many organs in addition to the lungs can be condemned (collected as samples if a study is undertaken), does anything similar exist in China, how is the slaughterhouse surveillance undertaken

- What are the last official prevalence data of bTB in cattle in China?

Line 50-54: add references

Line 239: you mention bTB nucleic acid, do you mean M. bovis or mycobacteria in general? please specify. Bovine tuberculosis is mostly caused by M. bovis, but other mycobacteria have been shown to cause the disease, so it's better to refer to the pathogen instead of the disease when talking about DNA. In addition, if you mean mycobacteria here, then please mention if you are talking about tuberculous or non tuberculous mycobacteria?

Table 2: I suggest you add brief details of the qPCR you used here (as mentioned above)

In the following study, performed in Ethiopia, the researchers found that the majority of bTB lesions were in the mesenteric lymph nodes, I would suggest to include this reference and also perform a deeper bibliographic search to supplement the discussion

1. Ameni G, Aseffa A, Engers H, Young D, Gordon S, Hewinson G, et al. High Prevalence and Increased Severity of Pathology of Bovine Tuberculosis in Holsteins Compared to Zebu Breeds under Field Cattle Husbandry in Central Ethiopia. Clin Vaccine Immunol. 2007 Oct 1;14(10):1356–61.

Line 208: Sections of all paraffin-embedded tissues isolated from ....

Line 259-260: I suggest you reword this sentence :

In several dairy farms of China, a high prevalence of bTB is reported by using the same method

Line 267: by screening 13,385.....

I suggest to english edit the manuscript by an english native speaker or a professional language editing service.

Reviewer #2: Several paragraphs need to be reviewed with regard to their style in English.

The authors should specify the conditions under which in vivo testing was conducted. Especially, if the animals have been subjected to parallel or serial testing. This is critical for result comparisons. According to International standards (OIE standard), two in vivo tuberculin tests for screening of bTB should be conducted 6 weeks apart at least . In addition the Eye drop reaction used by the authors is not considered as a reference test to screen for BTB. Published as such, would be misleading, inciting others researchers to use the eye drop reaction and to classify the reactors accordingly, as being in advanced stage of disease. Furthermore, the pathological post mortem findings in 9,16% of infected animals do not precise if the gastrointestinal associated lesions were primary or secondary foci. bTB is known to spread from primary foci (i.e. pulmonary...) to various tissues and organs via different routes. These informations should be provided to acertain data validity.

On the other hand, the provided ethical statement refers to technical standards of the Chinese competent authority and not to a university ethic committee.

Reviewer #3: In general terms, the manuscript shows a significant amount of information and reflects the great workload invested and the important investigation tools deployed.

The English style needs an improvement and homogenization. Reading by a native English speaker is recommended.

Description of the used screening tests did not mention their conditions of application and chronology (serial or parallel testing) on which the results interpretation is depending.

Objectives and purposes of using such a large number of screening tests have not been clearly defined, especially since one of the tests is not recognized as a reference test by international standards (PPD eye drop reaction). Furthermore, the authors did not specify how they selected animals with advanced infection based on test results.

Comparison of test results and the subsequent discussion will certainly bring an added value to this work, especially in terms of practical test use and related cost effectiveness.

6. PLOS authors have the option to publish the peer review history of their article (what does this mean?). If published, this will include your full peer review and any attached files.

Reviewer #1: No

Reviewer #2: No

Reviewer #3: No

---

## [Author Response · Author response to Decision Letter 0]

29 Oct 2020

Dear Prof. Quinn and Reviewers,

Thanks very much for your letter and for the reviewers’ comments concerning our manuscript entitled “High prevalence of extrapulmonary tuberculosis in dairy farms: evidence for possible gastrointestinal transmission” (ID: PONE-D-20-23226). I really appreciate all your comments and suggestions. We made revision as required with changes marked in red. We also made point-to-point response as highlighted in blue.

Replies to the editor:

Journal Requirements:

Response: The manuscript has revised according to the PLOS ONE style templates, including those for file naming.

2. In your Methods section, please provide the name of the slaughterhouse where the animals were sacrificed.

Response: Thanks. The names of the slaughterhouse where the animals were sacrificed were provided in Methods section (Lines 125-126, page 6).

3. Thank you for including your ethics statement: "All ante-mortem testing and post-mortem samples were obtained in compliance with national standards of the People's Republic of China (PRC): Diagnostic techniques for tuberculosis of animal (GB/T 18645-2002), and Bovine tuberculosis diagnosis-Assay on IFN-γ in vitro (GB/T 32945-2016); in accordance with national guidelines for the prevention and treatment of bovine tuberculosis (2017-2020) (Ministry of Agriculture and Rural Affairs of PRC); in accordance with legislation: Animal Epidemic Prevention Law of PRC (1997, order No.87 of president of PRC).".

Please amend your current ethics statement to confirm that your named ethics committee specifically approved this study.

For additional information about PLOS ONE submissions requirements for ethics oversight of animal work, please refer to http://journals.plos.org/plosone/s/submission-guidelines#loc-animal-research

Response: We are aware of the reviewer’s concern. In this epidemiological investigation, we detected different types of samples by using multiple methods, on the basis of the national standards, national guidelines and the legislation.

We have made correction and interpretation accordingly. (Lines 194-200, page 9-10).

“Ethical statement: Bovine tuberculosis (bTB) is a notifiable disease and there are control and surveillance campaigns in China. Official diagnostic methods for bTB are immunological tests, culture, PCR and histopathology. In this study, no animal experiment was involved. All datasets were in complete agreement with national and OIE regulations.”

Response: Thanks. We added that as S4 Fig.

Special thanks to you for your good comments.

Replies to the reviewers’ comments:

Reviewer #1:

1. Please explain from an immunological point of view how did you chose the different tests to design the early infection versus the advanced infection.

Response: We accept the reviewer’s critique and added the explanation in an immunological point of view (Lines 121-128, page 6).

2. I suggest you add the methods of each test briefly and then reference the instructions given by the vendor when appropriate. The same for the qPCR, it will be good to add a brief description of the experiment.

Response: As suggested, the methods of each test were briefly added in “Materials and Methods” section (Lines 89-117, page 5-6).

We added the part of “DNA extraction and purification” in “Materials and Methods” section (Lines 158-162, page 8).

We also added a brief description of the experiment to the qPCR (Lines 164-172, page 8).

3. In the methods section, I suggest you add a description of the livestock production systems (intensive or extensive) of the farms included in the study, in addition to a brief description of the animal housing conditions and the breeds of the cattle screened. Those are all elements that can influence the transmission and persistence of bTB in livestock.

Response: We added a description of the livestock production systems of the farms, the animal housing conditions and the breeds of the cattle screened (Lines 85-87, page 4-5).

4. Table 1 gives a great summary of the results of bTB diagnosis, however, it will be interesting to perform a multivariate analysis in order to detect any significant statistical difference between the different methods used.

Response: That’s a good point. Basically, bovine tuberculosis (bTB) is a chronic bacterial disease, the diagnosis of which is implicated to multiple methods during the whole course of infection. For instance, currently, three assays (SIT, CIT, and IFN-γ assay) that depended on the pro-inflammatory cell-mediated immune (CMI) response are generally performed for detection of cattle with early infection, while the ELISA that depended on humoral responses is somehow performed for detection of cattle with advanced infection. That is, at different periods of bTB infection, there are appropriate methods. We believe it is interesting to further focus on the limitations of these diverse methods at different periods of infection based on a large number of samples.

5. In addition, it will be useful to calculate the apparent and true prevalence considering the sensitivity and specificity of each test used (or the tests for which sensitivity and specificity have been previously determined).

Response: Thanks. As similar to Q4 that required the comparison among the different methods, at different periods of bTB infection, there are appropriate methods. Our further study may systematically focus on the true prevalence considering the sensitivity and specificity of each test based on true positive and negative samples.

6. I suggest to add some information to the introduction part, some questions to answer in the introduction:

- is the comparative tuberculin skin test using avian PPD in addition to bovine PPD used in China

- Are there any available molecular data of bTB in cattle in China (pulmonary and extra pulmonary)

- In Slaughterhouses, usually many organs in addition to the lungs can be condemned (collected as samples if a study is undertaken), does anything similar exist in China, how is the slaughterhouse surveillance undertaken

- What are the last official prevalence data of bTB in cattle in China?

Response: We added the information to the introduction part according to the Reviewer’s suggestion, and all the questions above have answered in the introduction (Lines 46-47, 56-58, 64-72, 76-78, page 3-4).

7. Line 50-54: add references.

Response: Thanks. The references have been added (Lines 53-55, page 3).

8. Line 239: you mention bTB nucleic acid, do you mean M. bovis or mycobacteria in general? please specify. Bovine tuberculosis is mostly caused by M. bovis, but other mycobacteria have been shown to cause the disease, so it's better to refer to the pathogen instead of the disease when talking about DNA. In addition, if you mean mycobacteria here, then please mention if you are talking about tuberculous or non tuberculous mycobacteria?

Response: Thanks. We modified the expression in these sentences according to the previous comment (Lines 292-297, page 14).

9. Table 2: I suggest you add brief details of the qPCR you used here (as mentioned above).

Response: Thanks. We added brief details of the qPCR in “Materials and Methods” section (Lines 164-172, page 8).

10. In the following study, performed in Ethiopia, the researchers found that the majority of bTB lesions were in the mesenteric lymph nodes, I would suggest to include this reference and also perform a deeper bibliographic search to supplement the discussion.

1. Ameni G, Aseffa A, Engers H, Young D, Gordon S, Hewinson G, et al. High Prevalence and Increased Severity of Pathology of Bovine Tuberculosis in Holsteins Compared to Zebu Breeds under Field Cattle Husbandry in Central Ethiopia. Clin Vaccine Immunol. 2007 Oct 1;14(10):1356–61.

Response: Thank you for the suggestion. We added this reference and the information required as explained above (Lines 345-353, page 16-17).

11. Line 208: Sections of all paraffin-embedded tissues isolated from ....

Response: We modified the sentence according to the previous comment (Line 262, page 13).

12. Line 259-260: I suggest you reword this sentence:

In several dairy farms of China, a high prevalence of bTB is reported by using the same method.

Response: We reworded the sentence according to the previous comment (Lines 316-317, page 15).

13. Line 267: by screening 13,385.....

Response: We modified the sentence according to the previous comment (Line 324, page 16).

14. I suggest to English edit the manuscript by an English native speaker or a professional language editing service.

Response: Thanks. We improved the language.

Special thanks to you for your good comments.

Reviewer #2:

1. Several paragraphs need to be reviewed with regard to their style in English.

Response: Thanks. We modified that.

2. The authors should specify the conditions under which in vivo testing was conducted. Especially, if the animals have been subjected to parallel or serial testing. This is critical for result comparisons. According to International standards (OIE standard), two in vivo tuberculin tests for screening of bTB should be conducted 6 weeks apart at least.

Response: We specified the conditions under which in vivo testing was conducted in “Materials and Methods” section (Line 101, page 5).

3. In addition the Eye drop reaction used by the authors is not considered as a reference test to screen for BTB. Published as such, would be misleading, inciting others researchers to use the eye drop reaction and to classify the reactors accordingly, as being in advanced stage of disease.

Response: Considering the Reviewer’s comments, we deleted the PPD eye drop reaction method and the related data which had no effect on the test results. We have corrected the Table 1 and S1 Fig.

4. Furthermore, the pathological post mortem findings in 9.16% of infected animals do not precise if the gastrointestinal associated lesions were primary or secondary foci. bTB is known to spread from primary foci (i.e. pulmonary...) to various tissues and organs via different routes. These informations should be provided to a certain data validity.

Response: We provided the information to a certain data validity according to the Reviewer’s comment (Lines 343-351, page 16-17).

5. On the other hand, the provided ethical statement refers to technical standards of the Chinese competent authority and not to a university ethic committee.

Response: We are aware of the reviewer’s concern. In this epidemiological investigation, we detected different types of samples by using multiple methods, on the basis of the national standards, national guidelines and the legislation.

We have made correction and interpretation accordingly. (Lines 196-200, page 9-10).

“Ethical statement: Bovine tuberculosis (bTB) is a notifiable disease and there are control and surveillance campaigns in China. Official diagnostic methods for bTB are immunological tests, culture, PCR and histopathology. In this study, no animal experiment was involved. All datasets were in complete agreement with national and OIE regulations.”

Special thanks to you for your good comments.

Reviewer #3: 

1. The English style needs an improvement and homogenization. Reading by a native English speaker is recommended.

Response: Thanks. We improved the language.

2. Description of the used screening tests did not mention their conditions of application and chronology (serial or parallel testing) on which the results interpretation is depending.

Response: We added the conditions of application and chronology of screening tests in “Materials and Methods” section (Lines 89-128, page 5-6).

3. Objectives and purposes of using such a large number of screening tests have not been clearly defined, especially since one of the tests is not recognized as a reference test by international standards (PPD eye drop reaction). Furthermore, the authors did not specify how they selected animals with advanced infection based on test results.

Response: Considering the Reviewer’s comments, we deleted the PPD eye drop reaction and the related data which had no effect on the test results. We have defined the objectives and purposes of using a large number of screening tests (Lines 119-121, page 6).

4. Comparison of test results and the subsequent discussion will certainly bring an added value to this work, especially in terms of practical test use and related cost effectiveness.

Response: We thank the comments.

Special thanks to you for your good comments.

Other changes:

1. Line 24-25: “8” was corrected as “eight”, “3” was corrected as “three”.

2. Line 30: “12” was corrected as “twelve”, “%” was corrected as “percent”.

3. Line 33: “10” was corrected as “ten”.

4. Line 35: “bTB” was corrected as “M. bovis”.

5. Line 59: “PPD” was corrected as “purified protein derivative (PPD)”.

6. Line 119: “Five” was corrected as “Four”, “8” was corrected as “eight”.

7. Line 174-175: “8” was corrected as “eight”, “6” was corrected as “six”, “4” was corrected as “four”.

8. Line 207: “8” was corrected as “eight”.

9. Line 238: “%” was corrected as “percent”.

10. Line 266: “mycobacterium tuberculosis complex” was corrected as “MTBC”.

11. Line 272-275: “7” was corrected as “seven”, “17” was corrected as “seventeen”, “10” was corrected as “ten”, “6” was corrected as “six”, “4” was corrected as “four”.

12. Line 279: “10” was corrected as “ten”.

13. Line 284: “17” was corrected as “seventeen”.

14. Line 293: “8” was corrected as “eight”.

15. Line 295: “12” was corrected as “Twelve”.

16. Line 297: “8” was corrected as “eight”.

17. Line 306: Table 2, “8” was corrected as “eight”.

18. Line 324: “4” was corrected as “four”.

19. Line 326: “8” was corrected as “eight”.

20. Line 329: “10” was corrected as “ten”.

Response: Thanks. We made revision.

---

## [Decision Letter · Decision Letter 1]

6 Jan 2021

PONE-D-20-23226R1

High prevalence of extrapulmonary tuberculosis in dairy farms: evidence for possible gastrointestinal transmission

PLOS ONE

Dear Dr. Zeng,

Thank you for submitting your manuscript to PLOS ONE. After careful consideration, we feel that it has merit but does not fully meet PLOS ONE’s publication criteria as it currently stands. Therefore, we invite you to submit a revised version of the manuscript that addresses the points raised during the review process.

Please submit your revised manuscript. If you will need significantly more time to complete your revisions, please reply to this message or contact the journal office at plosone@plos.org. Please include the following items when submitting your revised manuscript:

We look forward to receiving your revised manuscript.

Kind regards,

Frederick Quinn

Academic Editor

PLOS ONE

Reviewers' comments:

Reviewer's Responses to Questions

**Comments to the Author**

1. If the authors have adequately addressed your comments raised in a previous round of review and you feel that this manuscript is now acceptable for publication, you may indicate that here to bypass the “Comments to the Author” section, enter your conflict of interest statement in the “Confidential to Editor” section, and submit your "Accept" recommendation.

Reviewer #1: (No Response)

2. Is the manuscript technically sound, and do the data support the conclusions?

Reviewer #1: Yes

3. Has the statistical analysis been performed appropriately and rigorously? 

Reviewer #1: Yes

4. Have the authors made all data underlying the findings in their manuscript fully available?

Reviewer #1: Yes

5. Is the manuscript presented in an intelligible fashion and written in standard English?

Reviewer #1: No

6. Review Comments to the Author

Reviewer #1: Thank you for submitting a revised version and answering all the comments, the manuscript has improved so much, however, I still think it needs professional proofreading and english editing

One more question: did you collect the cervical lymph nodes and lung associated lymph nodes? if not, can you please explain why ? I'm asking this question because very often TB disease in cattle is contained in the lymph nodes and does not progress to other organs and tissues, so it is important to examine those specific lymph nodes for gross visible lesions (in addition to the other lymph nodes you mentioned you have examined)

Below few typos and suggestions

Line 29: were tested

Line 30: the results....had advanced infection of bTB

Line 34: replace of note by in fact

Line 38-39: the phylogenetic results... (reword this sentence please)

Line 60 The Single Intradermal Test (SIT) and IFN-γ assay are the current standard diagnostic

diagnosis for bTB in some cattle farms with good economic conditions: reword

Line 69: what do you mean by harmless treatment

Line 72: is frequently overlooked

Line 82: ...and extrapulmonary bTB....

Line 126 to 129: Reword this sentence

7. PLOS authors have the option to publish the peer review history of their article (what does this mean?). If published, this will include your full peer review and any attached files.

Reviewer #1: No

---

## [Author Response · Author response to Decision Letter 1]

18 Jan 2021

Dear Prof. Quinn and Reviewers,

Thanks very much for your letter and for the reviewers’ comments concerning our manuscript entitled “High prevalence of extrapulmonary tuberculosis in dairy farms: evidence for possible gastrointestinal transmission” (ID: PONE-D-20-23226R1). I really appreciate for all your comments and suggestions. We made revision as required with changes marked in red. We also made point-to-point response as highlighted in blue.

Replies to the reviewers’ comments:

Reviewer #1:

1. Did you collect the cervical lymph nodes and lung associated lymph nodes? if not, can you please explain why? I'm asking this question because very often TB disease in cattle is contained in the lymph nodes and does not progress to other organs and tissues, so it is important to examine those specific lymph nodes for gross visible lesions (in addition to the other lymph nodes you mentioned you have examined).

Response: That’s a good point. We examined lymph nodes and numerous organs and tissues in order to identify gross visible lesions, and we then collected all the organs and tissues with lesions, including the cervical lymph nodes and lung associated lymph nodes, such as submandibular lymph nodes and hilar lymph nodes.

2. Below few typos and suggestions.

Line 29: were tested

Line 30: the results....had advanced infection of bTB

Line 34: replace of note by in fact

Line 38-39: the phylogenetic results... (reword this sentence please)

Line 60 The Single Intradermal Test (SIT) and IFN-γ assay are the current standard diagnostic

diagnosis for bTB in some cattle farms with good economic conditions: reword

Line 69: what do you mean by harmless treatment

Line 72: is frequently overlooked

Line 82: ...and extrapulmonary bTB....

Line 126 to 129: Reword this sentence

Response: Thank you for the suggestions. We made modification according to the previous comments. We double checked the spelling and improved the language.

---

## [Decision Letter · Decision Letter 2]

3 Feb 2021

PONE-D-20-23226R2

High prevalence of extrapulmonary tuberculosis in dairy farms: evidence for possible gastrointestinal transmission

PLOS ONE

Dear Dr. Zeng,

Thank you for submitting your manuscript to PLOS ONE. After careful consideration, we feel that it has merit but does not fully meet PLOS ONE’s publication criteria as it currently stands. Therefore, we invite you to submit a revised version of the manuscript that addresses the points raised during the review process.

Please submit your revised manuscript. If you will need significantly more time to complete your revisions, please reply to this message or contact the journal office at plosone@plos.org. Please include the following items when submitting your revised manuscript:

We look forward to receiving your revised manuscript.

Kind regards,

Frederick Quinn

Academic Editor

PLOS ONE

Reviewers' comments:

Reviewer's Responses to Questions

**Comments to the Author**

1. If the authors have adequately addressed your comments raised in a previous round of review and you feel that this manuscript is now acceptable for publication, you may indicate that here to bypass the “Comments to the Author” section, enter your conflict of interest statement in the “Confidential to Editor” section, and submit your "Accept" recommendation.

Reviewer #1: All comments have been addressed

2. Is the manuscript technically sound, and do the data support the conclusions?

Reviewer #1: Yes

3. Has the statistical analysis been performed appropriately and rigorously? 

Reviewer #1: N/A

4. Have the authors made all data underlying the findings in their manuscript fully available?

Reviewer #1: Yes

5. Is the manuscript presented in an intelligible fashion and written in standard English?

Reviewer #1: No

6. Review Comments to the Author

Reviewer #1: Thank you for addressing all the comments, the manuscript has improved a lot, however, I still have few suggestions and questions listed below. In addition, I highly suggest detailed english editing of the manuscript, by an english native speaker or a professional editing service.

Line 30_31:

"The result indicated that 752 (5.64%) had advanced infection of bTB"

Line 39: delete "positive"

Line 41_42:

"Our data demonstrate that the increase of EPTB transmitted by digestive tract is implicated in the current high prevalence rate of bTB in China"

Line 50 : delete "concerne"

Line 61: "the comparative intradermal test (CIT) is also used often."

Line 62 to 64: here you mentioned studies showing that PPD positive cattle don't present typical Tb associated symptoms, and they don't possess lung lesions upon necropsy. Can you please reference those studies.

Line 65-66: "as well as interfered **with** the implementation of culling policy on the cattle suspected to be infected with bTB."

Line 132-133: I suggest to rewrite this part of the sentence as follow:

", were slaughtered and anatomical examination was performed in the local slaughterhouses named Kaerwan (Xinjiang), Musulin (Shandong), and Chengcheng (Guangxi)."

Line 139 anatomical examination was or anatomical examinations were

please modify accordingly

Line 150: "approximately"

Line 300-302: "Similar to milk samples, M. bovis nucleic acid was detected in all eight samples from fecal pool and from 34 positive cattle"

In the last sentence, which specific samples from 34 positive cattle are you talking about here? are they also fecal samples, or milk? please specify

7. PLOS authors have the option to publish the peer review history of their article (what does this mean?). If published, this will include your full peer review and any attached files.

Reviewer #1: No

---

## [Author Response · Author response to Decision Letter 2]

4 Mar 2021

Dear Prof. Quinn and Reviewers,

Thanks very much for your letter and for the reviewers’ comments concerning our manuscript entitled “High prevalence of extrapulmonary tuberculosis in dairy farms: evidence for possible gastrointestinal transmission” (ID: PONE-D-20-23226R2). We really appreciate all your comments and suggestions. We revised our manuscript, and we have sent the revised manuscript, and a version containing all the changes to be visible. We also made point-to-point response as highlighted in blue.

Replies to the reviewers’ comments:

Reviewer #1:

1. Below few typos and suggestions.

Line 30_31: "The result indicated that 752 (5.64%) had advanced infection of bTB"

Line 39: delete "positive"

Line 41_42: "Our data demonstrate that the increase of EPTB transmitted by digestive tract is implicated in the current high prevalence rate of bTB in China"

Line 50: delete "concerne"

Line 61: "the comparative intradermal test (CIT) is also used often."

Line 65-66: "as well as interfered **with** the implementation of culling policy on the cattle suspected to be infected with bTB."

Line 132-133: I suggest to rewrite this part of the sentence as follow:

", were slaughtered and anatomical examination was performed in the local slaughterhouses named Kaerwan (Xinjiang), Musulin (Shandong), and Chengcheng (Guangxi)."

Line 139: anatomical examination was or anatomical examinations were

please modify accordingly

Line 150: "approximately"

Response: Thank you for the suggestions. We modified and reworded the sentences according to the previous comment.

2. Line 62 to 64: here you mentioned studies showing that PPD positive cattle don't present typical Tb associated symptoms, and they don't possess lung lesions upon necropsy. Can you please reference those studies.

Response: Thanks. We had listed the references behind the sentence.

3. Line 300-302: "Similar to milk samples, M. bovis nucleic acid was detected in all eight samples from fecal pool and from 34 positive cattle"

In the last sentence, which specific samples from 34 positive cattle are you talking about here? are they also fecal samples, or milk? please specify

Response: Thank you for the suggestion. We modified the sentences according to the previous comment. In the last sentence, we had specified the 34 feces samples from bTB-positive cattle.

4. In addition, I highly suggest detailed english editing of the manuscript, by an english native speaker or a professional editing service.

Response: Thank you for the suggestion. We modified other spelling and improved the language. English expression has been carefully improved throughout the manuscript. And we made a marked-up copy of manuscript that highlights changes to the original version named 'Revised Manuscript with Track Changes'.

Special thanks to you for your good comments.

Thanks. We made revision.

---

## [Decision Letter · Decision Letter 3]

17 Mar 2021

High prevalence of extrapulmonary tuberculosis in dairy farms: evidence for possible gastrointestinal transmission

PONE-D-20-23226R3

Dear Dr. Zeng,

We’re pleased to inform you that your manuscript has been judged scientifically suitable for publication and will be formally accepted for publication once it meets all outstanding technical requirements.

Kind regards,

Frederick Quinn

Academic Editor

PLOS ONE

Additional Editor Comments (optional):

Reviewers' comments:

Reviewer's Responses to Questions

**Comments to the Author**

1. If the authors have adequately addressed your comments raised in a previous round of review and you feel that this manuscript is now acceptable for publication, you may indicate that here to bypass the “Comments to the Author” section, enter your conflict of interest statement in the “Confidential to Editor” section, and submit your "Accept" recommendation.

Reviewer #1: All comments have been addressed

2. Is the manuscript technically sound, and do the data support the conclusions?

Reviewer #1: Yes

3. Has the statistical analysis been performed appropriately and rigorously? 

Reviewer #1: Yes

4. Have the authors made all data underlying the findings in their manuscript fully available?

Reviewer #1: Yes

5. Is the manuscript presented in an intelligible fashion and written in standard English?

Reviewer #1: Yes

6. Review Comments to the Author

Reviewer #1: Following few suggestions and typos:

Line 72: become the most.... (delete as)

Line 80-81: reword this sentence please

Line 114: ...immune responses can facilitate the detection if late stage....

Line 173-178: cattle is the plural form please replace by the singular form, which is the age or sex-specific terms of the animal (e.g: cow, bull....)

7. PLOS authors have the option to publish the peer review history of their article (what does this mean?). If published, this will include your full peer review and any attached files.

Reviewer #1: No

---

## [Editor Report · Acceptance letter]

22 Mar 2021

PONE-D-20-23226R3 

High prevalence of extrapulmonary tuberculosis in dairy farms: evidence for possible gastrointestinal transmission 

Dear Dr. Zeng:

I'm pleased to inform you that your manuscript has been deemed suitable for publication in PLOS ONE. Congratulations! Your manuscript is now with our production department. 

Kind regards, 

on behalf of

Dr. Frederick Quinn 

Academic Editor

PLOS ONE